# Improving Toilet Usability and Cleanliness in Public Schools in the Philippines Using a Packaged Operation and Maintenance Intervention

**DOI:** 10.3390/ijerph191610059

**Published:** 2022-08-15

**Authors:** Denise Duijster, Bella Monse, Marvin Marquez, Ubo Pakes, Nicole Stauf, Habib Benzian

**Affiliations:** 1Department of Oral Public Health, Academic Center for Dentistry Amsterdam (ACTA), University of Amsterdam and VU University Amsterdam, Gustav Mahlerlaan 3004, 1081 LA Amsterdam, The Netherlands; 2Regional Fit for School Programme, Deutsche Gesellschaft für Internationale Zusammenarbeit (GIZ), 10th Floor, Bank of Makati Building, Ayala Avenue Extension near Corner Metropolitan Avenue, Makati City 1209, Philippines; 3Population Institute, College of Social Science and Philosophy, University of the Philippines Diliman, Diliman, Quezon City 1101, Philippines; 4Center for Environmental Informatics, University of the Philippines Cebu, Gorordo Avenue, Cebu City 6000, Philippines; 5The Health Bureau Ltd., Whiteleaf Business Center, 11 Little Balmer, Buckingham MK18 1TF, UK; 6WHO Collaborating Center Quality Improvement & Evidence-Based Dentistry, Department of Epidemiology & Health Promotion, College of Dentistry, New York University, 433 First Avenue, 7th Floor, New York, NY 10010, USA

**Keywords:** Water, Sanitation and Hygiene (WASH), operation and maintenance, toilets, sanitation, schools, cleanliness

## Abstract

This study evaluated the impact of packaged interventions for operation and maintenance (O&M) on the usability and cleanliness of toilets in public schools in the Philippines. In this cluster-randomized controlled trial, the divisions of Roxas City and Passi City were randomly assigned to the intervention or control group. Schools in Roxas City (n = 14) implemented the packaged O&M interventions; schools in Passi City (n = 16) formed the control group. Outcome variables were toilet usability—defined as accessible, functional and private—and toilet cleanliness, measured using the Sanitation Assessment Tool (SAT) and the Cleaner Toilets, Brighter Future (CTBF) instruments at baseline and at four months follow-up through direct observation of school toilets. SAT results showed that intervention schools had a 32.0% (4.6%; 59.3%) higher percentage of usable toilets than control schools at follow-up after full adjustment (*p* = 0.024). CTBF results found a similar result, although this was not statistically significant (*p* = 0.119). The percentage of toilets that were fully clean was 27.1% (3.7%; 50.6%) higher in intervention schools than in control schools after adjustment (*p* = 0.025). SAT results also showed an improvement in cleanliness of toilets in intervention schools compared to those in controls, but this did not remain significant after adjustment. The findings indicate that the additional implementation of O&M interventions can further stimulate progress towards reaching Water, Sanitation and Hygiene service levels aligned with the Sustainable Development Goals.

## 1. Introduction

Sustainable access to Water, Sanitation and Hygiene (WASH) services in schools is essential to promote health and positive educational outcomes of children [1]. In Southeast Asia, including the Philippines, many children suffer from a high burden of preventable diseases with inadequate sanitation, limited access to water and lack of personal hygiene as a common cause [2,3,4]. These diseases, such as diarrhea, respiratory infections and soil-transmitted helminth (STH) infections, negatively affect children’s overall development through their adverse impact on school attendance, educational achievement and quality of life [5,6]. Children spend a substantial amount of their time at school where they benefit from a safe and healthy learning environment with adequate WASH facilities and routine activities fostering positive hygiene habits. However, many schools are facing challenges with providing and maintaining even basic service levels as defined in the Sustainable Development Goals (SDGs). Common bottlenecks include the lack of resources and weak management structures to ensure regular operation and maintenance (O&M) of WASH facilities [7,8,9]. As a result, many toilets remain unusable, are not kept clean and/or basic requirements for pupil handwashing are not met.

WASH in schools (WinS) is a global movement to improve WASH services in the school setting. WinS is anchored in the SDGs under goal 4 (“Education for all”) which targets to achieve access to drinking water, basic sanitation and handwashing facilities with soap in all schools by 2030 [10]. The World Health Organization/UNICEF Joint Monitoring Programme for Water Supply, Sanitation and Hygiene (JMP) developed a framework of service levels to support countries on their way to reach the WASH-related SDG indicators [11]. The service levels include four categories: no service, limited service, basic service and advanced service, with the basic service level describing the level of service required to meet the SDG objective. Basic sanitation service refers to toilet facilities in schools that are single-sex and usable, meaning they are accessible (doors are unlocked or the key is always available), functional (toilet is not broken, hole is unblocked and water is available for flushing) and private (doors can be locked from the inside and users are protected from outside views). An additional criterium for advanced sanitation service includes cleanliness. Regular cleaning of toilets is a key element in reducing toilet avoidance; a common problem among pupils that has been associated with decreased concentration and a higher risk of urinary infections, bladder and bowel disturbances, STH infections and reduced fluid intake [12,13].

To strengthen WinS in Southeast Asia, the Deutsche Gesellschaft für Internationale Zusammenarbeit (GIZ) and the Southeast Asian Ministers of Education Organization (SEAMEO) are joining forces to support Ministries of Education to improve WinS programming, using the Fit for School (FIT) approach [14,15]. The FIT approach is an integrated school health concept, bringing together WASH improvements and effective health interventions as part of routines in public schools and within the management responsibilities of school heads and teachers. The approach aims to improve health and learning conditions of school children by supporting Ministries of Education with the institutionalization of simple, evidence-based and cost-effective interventions in schools, including the construction of group handwashing facilities, concepts for operation and maintenance of sanitation facilities, the integration of daily group handwashing with soap and toothbrushing with fluoride toothpaste and biannual deworming.

The Department of Education (DepEd) in the Philippines has integrated the FIT approach into its national school health programming as the Essential Health Care programme since 2009 [15]. Evaluation studies have shown that the approach was associated with improved child health outcomes and increased access to handwashing facilities [16,17]. However, school-level capacity to clean and maintain school toilets specifically remained a barrier to improving the situation at scale. In response to this, an additional package for WinS O&M was developed in 2016, targeting to increase the percentage of school toilets that are usable and clean [18]—in alignment with the SDG and related JMP definitions for service levels. The O&M package aims to support school staff to conduct regular O&M of sanitation facilities through clear definition and costing of supplies, spare parts and tools as well as information such as guidance manuals, respective budget allocation needed and practical tools for cleaning. Following a first evaluation of the concept, the packaged O&M concept has been included in the implementing guidelines for WinS in 2019. The aim of the current study was to evaluate the impact of the packaged O&M interventions on the usability and cleanliness of toilets through a cluster-randomized trial in public schools in the Philippines. In this study, two new instruments to measure toilet usability and cleanliness were used and compared.

## 2. Materials and Methods

### 2.1. Study Location and Context

This study was conducted in Roxas City and Passi City on Panay Island, the Philippines. The study included public elementary schools, secondary schools and integrated schools (offering kindergarten, elementary and high school education). Orienting visits to schools in the area prior to the study revealed that toilets in Roxas City and Passi City were predominantly flush or pour flush toilets, connected to an on-site septic system. Available water was often piped water, or a combination of piped water, rainwater or water from a deep well. The water could not be used for drinking. Water containers or buckets with taps and a dipper were often available inside the toilet cubicle for flushing and/or anal cleansing. Toilet paper was not available in public schools. Toilets were located on school premises or in classrooms. Those on school premises were generally single-sex toilets and organized into blocks of two to three individual cubicles. Most block toilets had one or multiple sinks or basins with a tap available in a washroom in front of toilet cubicles for handwashing. Classroom toilets were not sex-segregated and were located within the classroom with a sink immediately outside the toilet entrance in the classroom.

### 2.2. Study Design

In this cluster-randomized controlled trial, the packaged interventions were delivered at the division level. The divisions of Roxas City and Passi City were randomly assigned to the intervention or control arm by flipping a coin. Schools in Roxas City implemented the packaged O&M interventions, while schools in Passi City formed the control group and received the interventions later, at the end of the study period. The interventions were delivered and evaluated over a four-month period from the first week of November 2019 to the first week of March 2020. The initial design was to perform an evaluation assessment after four and eight months, but due to the closing of schools mid-March 2020 as a result of the COVID-19 pandemic, the eight-month assessment was cancelled.

### 2.3. Study Sample

The research team requested DepEd to identify schools in the Roxas City and Passi City divisions which met the following inclusion criteria: school population of more than 300 students (to ensure sufficient school toilets), accessible location (within two hours from the city centre and with a mobile phone signal), secure with stable terrain for vehicle access and access to a water source. Thirty-seven schools met the inclusion criteria: 20 schools in Roxas City and 17 schools in Passi City. Previous studies on the FIT programme [17,18] have shown that a sample of 20 schools provides sufficient power to show statistically significant differences in toilet conditions between intervention and control schools.

### 2.4. The Interventions

The packaged O&M interventions were developed in a previous project [18] aligned with DepEd’s national guidelines for WinS and further improved based on valuable learnings and experiences from end-users. The package was developed to support school heads, staff and WinS coordinators with the O&M of WinS facilities, with particular focus on practical approaches on how to improve toilet conditions in schools. Schools in the intervention arm received material kits for toilet O&M, including a user’s kit (toilet brushes, buckets, dippers, locks and a trash bin with a cover for each toilet cubicle), a cleaner’s kit (mops, buckets, brooms, sponges, gloves and bleach) and a maintainer’s kit (with essential tools such as a screwdriver, hammer, a plunger and extra faucets). In addition, intervention schools received a detailed manual on toilet O&M that included WinS monitoring worksheets, budget allocation guidelines, example cleaning rotas and checklists [19].

At the start of the study, school staff in intervention schools received an orientation training by the research team during which they received the manual and materials with instructions on how to use them. School staff also received a video-based orientation training which was produced by DepEd and GIZ in the context of a Massive Open Online Course (MOOC) for school staff to strengthen O&M of school sanitation. After every two to three weeks, a member of the research team visited the schools to provide new consumables (soap and cleaning supplies). During these visits, the research team member informally monitored progress with the implementation of the packaged interventions and asked school staff if they needed any technical assistance. The manual and video were both available in English and Tagalog and are made available as Appendix A.

### 2.5. Outcome Variables and Instruments

Outcome variables of the study were (1) toilet usability—defined as accessible, functional and private—and (2) toilet cleanliness. Two instruments were used to measure toilet usability and cleanliness, namely the Sanitation Assessment Tool (SAT) and the Cleaner Toilets, Brighter Future (CTBF) instrument [20]. Both instruments are made available as Appendix A.

The SAT was developed by the FIT research team with the aim to enable simple and valid assessment of sanitation conditions in schools. A modified version of the UNICEF WinS monitoring tool [21] and the JMP Recommended Core Questions [22] served as a basis for the SAT development. The SAT was tested and refined through several pilot runs in two large elementary schools in Quezon City, Manila, Philippines. School principals and one member of the research team collected data using the SAT, after which debriefing meetings were held to evaluate the understandability, feasibility and content validity of the SAT and to identify suggestions for refinement.

The SAT assesses toilets on four aspects. First, the examiner scores whether a toilet is a girls’ toilet, a boys’ toilet, a shared toilet or a urinal. Second, the examiner evaluates whether a toilet is accessible and functional. A toilet is scored with a “yes” if it is currently in use (meaning not locked from the outside) and if the toilet hole is not blocked and water is available for flushing. Third, the privacy of those toilets that are accessible and functional is assessed by scoring whether the toilet can be locked from the inside. Lastly, the cleanliness of those toilets that are accessible, functional and private is scored into three categories: “clean”, “somewhat clean” and “not clean”. Cleanliness is determined by the absence of a strong smell, significant numbers of flies and insects and visible fecal matter in the toilet or around the facility. Each toilet in the school is assessed and scored by inserting tallies on a scoring sheet. From the tallies, the following school-level variables can be computed: the number and percentage of (i) toilets that are sex-segregated, (ii) toilets that are accessible and functional, (iii) toilets that are usable and (iv) usable toilets that are clean, somewhat clean and not clean.

The second assessment tool is the CTBF tool, which was designed to provide an overview of the sanitation status in schools. The tool was developed as part of the Unilever and Domestos CTBF programme to guide schools with planning O&M and improvements of school toilets. The concept of CTBF is aligned with the FIT approach and enhanced by the learnings from a public–private partnership between Unilever and GIZ focusing on improving usable and clean toilets through O&M. The simple tool consists of ten questions covering aspects of functionality, accessibility, privacy and cleanliness and based on the core and expanded questions of the JMP. Response options are limited to “yes” or “no”. Five questions assess toilet usability, including whether (i) the toilet cubicle has a door, (ii) the toilet door can be opened (is unlocked), (iii) the toilet cubicle protects the user from outside views, (iv) the toilet door can be locked from the inside and (v) the toilet bowl is intact. The other five questions assess toilet cleanliness including whether (vi) the toilet cubicle is free of litter, (vii) the toilet hole in unblocked, (viii) water for flushing is available inside the cubicle, (ix) the toilet bowl/pan/slab is free from visible traces of feces and urine and (x) the cubicle walls and floor are free of visible traces of feces and urine. For each toilet in the school, the number of criteria met is computed which can range between zero and ten.

### 2.6. Data Collection Procedure

Visual inspection surveys were conducted in each school at baseline and four months after implementation of the packaged O&M interventions. Data collection involved direct observation of non-classroom (block) toilets. Classroom toilets were excluded from the assessment because they are usually in better condition than non-classroom (block) toilets [18], which may dilute the findings of the intervention’s effect. Visual inspections were performed during one day shift (early morning, late morning, early afternoon or late afternoon) by a team of three external examiners who were trained and calibrated by a member of the research team prior to data collection. During data collection, one examiner labelled the toilet, one examiner assessed the toilet using the SAT and one examiner using the CTBF instrument. They were blind to the intervention status of the schools and tasks were randomly rotated among the three examiners per school. Data were collected on paper and using hand-held digital devices enabled through KoBo Toolbox, a digital platform for data collection. Checks were performed at the end of each data collection round for consistencies between the number of toilets assessed using the SAT and CTBF tool.

### 2.7. Final Selection of Schools

The baseline survey conducted in all 37 schools revealed that four schools in Roxas City and three schools in Passi City had excellent toilet conditions. Toilets in these schools were usable and clean according to the SAT and toilets met between 9 and 10 CTBF criteria. Since there was no opportunity to further improve toilet conditions by testing the packaged O&M interventions, these schools were excluded from the study. This resulted in a sample of 16 intervention schools in Roxas City and 14 control schools in Passi City.

### 2.8. Statistical Analysis

Data were analyzed using STATA v.16 (StataCorp, College Station, TX, USA). Toilet conditions, including mean percentages of toilets per school meeting the SAT and CTBF criteria, were compared between intervention and control schools at baseline and follow-up using the Mann–Whitney U-test. Differences in toilet conditions between baseline and follow-up in each group were analyzed using the Wilcoxon test. Linear regression was performed using a difference in difference analysis approach to provide the mean difference in the percentage of toilets meeting the SAT or CTBF criteria in intervention schools compared to control schools, after adjusting for differences at baseline. Since school toilets are usually being cleaned in the early mornings of a school day, the regression analyses were adjusted for the day shift when data collection was undertaken (early morning, late morning, early afternoon or late afternoon) to account for potential differences in the time between the last cleaning of school toilets and the assessment. A *p*-value of <0.05 was regarded as statistically significant.

## 3. Results

The characteristics of study schools are shown in Table 1. The study sample included 16 elementary schools, nine secondary schools and five integrated schools. Nine elementary schools, three secondary schools and four integrated schools were located in Roxas City, and seven elementary schools, six secondary schools and one integrated school were based in Passi City. All schools had access to water. Toilet conditions did not significantly differ between elementary, secondary and integrated schools at baseline. Intervention schools had a significantly lower mean number of children per school (662 ± 381) compared to that in control schools (894 ± 1199) (*p* < 0.001). Hence, the mean number of non-classroom (block) toilets was also lower in intervention schools than in control schools, both at baseline and follow-up, but this was not statistically significant (*p* = 0.067). At baseline, toilet conditions in terms of usability were comparable between intervention and control schools, yet in terms of cleanliness, toilets conditions were better on average in intervention schools (Table 2 and Table 3). Particularly, the percentage of usable toilets that were clean or somewhat clean was significantly higher in intervention than in control schools (*p* = 0.019) (Table 2).

At follow-up, there were on average two more toilets per school assessed compared to baseline, in both intervention and control schools. The main reasons for the increase relate to unclogging and unlocking toilets with a blocked hole, and opening toilets for use that were permanently closed or used for storage. Over 80% of non-classroom toilets in both intervention and control schools were sex-segregated.

### 3.1. SAT Results

The SAT results are shown in Table 2. The mean percentage of toilets that were accessible and functional in intervention schools significantly increased between baseline and follow-up (*p* = 0.004). In control schools, a smaller, non-significant increase was observed. At follow-up, the mean percentage of accessible and functional toilets was 11.0% (−13.6%; 35.6%) higher in intervention schools than in control schools after full adjustment. However, this difference was not statistically significant (*p* = 0.368). When including the aspect of privacy, the mean percentage of usable (accessible, functional and private) toilets in intervention schools significantly increased between baseline and follow-up (*p* = 0.002). In control schools, no difference was observed. At follow-up, intervention schools had a 32.0% (4.6%; 59.3%) higher percentage of usable toilets than control schools after full adjustment (*p* = 0.024). The percentage of usable toilets that were “clean” or “clean and somewhat clean” was already higher in interventions schools at baseline. A small increase in the cleanliness of intervention toilets was observed between baseline and follow-up (which was statistically significant for the percentage of clean and somewhat clean toilets), yet no change was seen in the cleanliness of toilets in control schools. The adjusted mean difference in the percentage of “clean” and “clean or somewhat clean” toilets did not significantly differ between intervention and control schools.

### 3.2. CTBF Results

The CTBF results are shown in Table 3. At baseline, toilets met an average of 5.5 ± 2.4 and 6.2 ± 2.3 CTBF criteria in intervention and control schools, respectively. At follow-up, this number significantly increased to 8.0 ± 2.4 CTBF criteria per toilet in intervention schools (*p* = 0.001), while no significant change was observed in control schools. At follow-up, the mean number of CTBF criteria met per toilet was 2.8 (1.0; 4.6) higher in intervention schools than in control schools after full adjustment (*p* = 0.003).

The percentage of toilets that met all ten CTBF criteria significantly increased in intervention schools between baseline and follow-up (*p* = 0.001), while no change was seen in control schools. At follow-up, the percentage of toilets meeting all ten CTBF criteria was 45.9% (29.3%; 62.5%) higher in intervention schools compared to that in control schools after full adjustment (*p* < 0.001). When looking at the five toilet usability criteria only, the percentage of intervention schools meeting all five usability criteria significantly increased from 19.0% at baseline to 63.0% at follow-up (*p* < 0.001), while this percentage decreased in control schools. At follow-up, the percentage of intervention schools meeting all five usability criteria was 24.8% (6.8%; 56.5%) higher than in control schools after full adjustment, but this was not statistically significant (*p* = 0.119). However, with regard to cleanliness, the percentage of toilets that met all five cleanliness criteria was 27.1% (3.7%; 50.6%) higher compared to control schools after adjustment, which was statistically significant (*p* = 0.025).

## 4. Discussion

To strengthen school-level capacity to provide and maintain basic sanitation service levels, as defined by the SDGs, this study evaluated the impact of packaged O&M interventions on toilet conditions in public schools in the Philippines. The findings showed that the packaged interventions, including material and supplies (user’s, cleaner’s and maintainer’s kits), information and management support tools (detailed manual, worksheets and checklists) and training (orientation video), were associated with significant improvements in the usability and cleanliness of school toilets after four months.

The percentage of accessible and functional toilets in intervention schools significantly increased by 11% in comparison to that in control schools (based on the SAT results), which could be attributed to more toilets being repaired or unclogged due to the packaged O&M interventions. However, an increase in the number of accessible and functional toilets was observed in both intervention and control schools. This may have been the direct consequence of the baseline assessment, sparking awareness of toilet conditions among school staff and insights into where there was room for improvement. Unclogging toilets and opening toilets that were permanently closed or used for storage would have been “low hanging fruit” to improve the accessibility and functionality of their school toilets. The hypothesis that monitoring or assessment of toilet conditions in itself might already be a driver for change is supported by data showing that more schools are reaching basic WASH service levels after the introduction of an annual WinS monitoring system in many countries across the globe [11,23]. Another factor that could have played a role is the Hawthorne effect [24], meaning that schools improved toilet conditions in response to the awareness of being observed in the context of this research.

In terms of the usability of toilets (including accessibility, functionality and privacy), this study showed that most improvements were seen in the increased privacy of toilets in intervention schools. When including the aspect of privacy, SAT results showed that intervention schools had a 32.0% higher percentage of usable toilets than control schools did at follow-up after full adjustment. This may indicate that the simple provision of locks as part of the user’s kit, in combination with instructions and guidance, is an effective measure to improve usability of toilets that are already accessible and functional. Ensuring private sanitation facilities is a generally important step to provide safety, dignity and wellbeing of all school children, and specifically supports girls attending school during menstruation [25].

Conflicting findings were observed regarding the interventions’ effects on toilet cleanliness when comparing the SAT results versus the CTBF results. The SAT results showed a small significant improvement in the percentage of “clean” and “somewhat clean” toilets in intervention schools, and not in control schools, but overall, no statistical differences in toilet cleanliness between intervention and control schools were detected at follow-up. Reasons for this include that the SAT uses a relatively crude measure to assess cleanliness, which makes it more difficult to show subtle differences in toilet cleanliness between groups. Additionally, the percentage of “clean” toilets (72.6%) and “clean” and “somewhat clean toilets” (92.1%) in intervention schools was already much higher at baseline, leaving less room for intervention schools to improve in comparison to control schools. The CTBF results, on the other hand, showed a clear significant improvement in toilet cleanliness conditions, with a 27.1% increase in toilets meeting all five CTBF cleanliness criteria compared to those in control schools. One explanation for this contrasting finding is that the CTBF assesses cleanliness in more detail, using multiple indicators that are less prone to subjective interpretation. Particularly, the indicators “the toilet cubicle is free of litter”, “the toilet bowl/pan/slab is free from visible traces of feces and urine” and “water for flushing is available inside the cubicle” were improved between baseline and follow-up in intervention schools (results not shown in a table). This may be a likely result from the provision of the cleaner’s kit in combination with the manual, cleaning rotas and checklist and the O&M orientation video.

A previous trial on the effectiveness of the packaged O&M interventions, conducted in Batangas, Philippines, found no significant impact on toilet usability, yet an increase in students’ satisfaction with sanitation facilities was observed [18]. In the Batangas study, variability in toilet conditions between schools was limited, and conditions at baseline were better; a different instrument to measure sanitation conditions was used (the Toilet Usability Index); and compliance with the intervention requirements was not optimal, which may explain the differences in findings between the Batangas trial and the current study. Other previous intervention studies have demonstrated that (a combination of) the provision of handwashing facilities and consumables (soap and cleaning supplies), budget allocation exercises, school-based management training, presence of a local champion, student and parental monitoring and involvement and support of the community are all effective components to improve sanitation O&M [8,9,26,27,28,29,30,31,32]. The packaged O&M interventions at focus in this study aimed to structurally improve toilet infrastructure by providing schools with consumables, supplies and guidance for toilet maintenance, as well as tools to identify needed budget for repairs. Furthermore, the packaged O&M interventions were implemented during a time when the national WinS programme in the Philippines already existed. In this programme, capacity building, regular monitoring, provision of technical assistance by WinS coordinators at division level and community support are taking place. These aspects together likely contributed to the observed improvements in toilet conditions in this study.

There are a number of strengths and limitations of this study that should be taken into account in the interpretation of findings. One strength is the use of two instruments based on the JMP Recommended Core Questions to measure toilet conditions; both were appropriate, simple and clear in use and have discriminative power. Both instruments, however, provide different perspectives on the usability and cleanliness of toilets. For example, the SAT uses a stepwise assessment, distinguishing between toilets that are only “accessible” and “functional”, and those that are also “private”, and in addition, “clean”, while the CTBF provides more detail about which aspects of toilet usability and cleanliness are being met. A limitation of the study includes the difference in toilet cleanliness at baseline, which could be the result of the cluster randomization. Since schools were randomized at the division level (two clusters) and not at the school level to avoid contamination of the intervention, school conditions were not fully comparable between intervention and control schools. Furthermore, schools in Region 6 of the Philippines, in which Roxas City and Passi City are located, are actively participating in the annual WinS monitoring (100% participation compared to 88% national participation in 2019/2020), and WASH services in schools are relatively positive compared to the national average. The included schools may therefore not be fully representative of public schools in other regions in the Philippines, particularly in those where participation in WinS monitoring is lower. Compliance to the packaged O&M interventions, referring to the extent to which schools followed recommended procedures and implemented activities, was not included in the study, which could be an important moderator of the intervention’s effect. Lastly, it should be noted that the improvements observed were achieved in a short four-month follow-up period, which is a remarkable finding in itself and shows that even with simple interventions and little training input, improvements are possible. It would be interesting for future research to revisit schools when they re-open to evaluate the long-term impact and whether the effects have been maintained beyond the short observation period.

## 5. Conclusions

This cluster-randomized controlled trial found that the implementation of packaged O&M interventions could improve toilet conditions in public schools in the Philippines over a period of four months. The combination of providing material and supplies for using, cleaning and structural maintenance of toilets, as well as information and training to guide the implementation of O&M activities in schools, was associated with improved usability and cleanliness of school toilets. Further research is needed to draw conclusions on the long-term impact of the packaged O&M interventions. The national WinS monitoring system, implemented by DepEd since 2017, has been an important step to support the education sector and schools to identify areas for WinS improvement and to facilitate sector planning and management. This study indicates that the additional implementation of the packaged O&M interventions, as part of the DepEd’s WinS programming in the Philippines, may be able to further stimulate progress towards reaching WASH service levels aligned with the SDG objectives.

## Figures and Tables

**Table 1 ijerph-19-10059-t001:** Characteristics of intervention and control schools at baseline and follow-up.

	Baseline		Follow-Up	
	Intervention Schools	Control Schools	Intervention Schools	Control Schools
	Mean ± sd	Mean ± sd	Mean ± sd	Mean ± sd
Number of boys’ toilets	3.1 ± 2.4	3.8 ± 5.2	3.6 ± 4.3	4.4 ± 8.2
Number of girls’ toilets	3.5 ± 2.9	4.6 ± 5.9	4.6 ± 6.2	4.9 ± 8.4
Number of shared toilets	1.2 ± 1.6	1.5 ± 2.3	1.6 ± 1.4	0.9 ± 1.6
Total number of toilets	7.8 ± 5.5	9.9 ± 12.5	10.1 ± 12.3	12.9 ± 20.1

**Table 2 ijerph-19-10059-t002:** Percentage of toilets meeting the SAT criteria in intervention and control schools at baseline and follow-up.

	Baseline			Follow-Up			
	Intervention Schools	Control Schools		Intervention Schools	Control Schools		
	Mean% ± sd	Mean% ± sd	*p* *	Mean% ± sd	Mean% ± sd	*p* *	Mean Difference β (95% CI), *p* **
Percentage of toilets that are accessible and functional	59.2 ± 30.4	59.0 ± 36.3	0.886	82.9 ± 28.1 ^1^	71.7 ± 34.6	0.224	11.0 (−13.6; 35.6), 0.368
Percentage of toilets that are usable (accessible, functional and private)	36.7 ± 28.6	35.0 ± 30.1	0.905	72.9 ± 28.1 ^2^	33.5 ± 31.1	0.003	32.0 (4.6; 59.3), 0.024
Percentage of usable toilets that are clean	72.6 ± 29.0	52.9 ± 35.3	0.126	76.4 ± 31.3	48.8 ± 38.8	0.047	2.1 (−29.9; 34.2), 0.893
Percentage of usable toilets that are clean and somewhat clean	92.1 ± 13.7	68.8 ± 27.7	0.019	99.1 ± 2.6 ^3^	70.4 ± 43.5	0.029	−2.0 (−21.5; 17.5), 0.834

* Mann–Whitney U-test, ** linear regression model: DID (difference in difference), adjusted for day shift. ^1^ Significantly different from baseline, *p* = 0.004 (Wilcoxon test). ^2^ Significantly different from baseline, *p* = 0.002 (Wilcoxon test). ^3^ Significantly different from baseline, *p* = 0.043 (Wilcoxon test).

**Table 3 ijerph-19-10059-t003:** Percentage of toilets meeting the CTBF criteria and mean CTBF scores of toilets in intervention and control schools at baseline and follow-up.

	Baseline			Follow-Up			
	Intervention Schools	Control Schools		Intervention Schools	Control Schools		
	Mean ± sd	Mean ± sd	*p* *	Mean ± sd	Mean ± sd	*p* *	Mean Difference β (95% CI), *p* **
CTBF score of toilets (mean number of criteria met)	5.5 ± 2.4	6.2 ± 2.3	0.394	8.0 ± 2.4 ^1^	6.0 ± 2.5	0.017	2.8 (1.0; 4.6), 0.003
Percentage of toilets meeting all 10 CTBF criteria	13.5 ± 20.1	15.7 ± 19.8	0.583	55.0 ± 33.6 ^2^	11.9 ± 17.3	<0.001	45.9 (29.3; 62.5), <0.001
Percentage of toilets meeting all 5 usability criteria ^†^	19.0 ± 25.8	32.0 ± 32.2	0.229	63.0 ± 29.3 ^3^	16.7 ± 20.5	<0.001	24.8 (6.8; 56.5), 0.119
Percentage of toilets meeting all 5 cleanliness criteria ^‡^	41.7 ± 23.1	29.6 ± 24.9	0.181	69.8 ± 31.4 ^4^	30.1 ± 33.0	0.005	27.1 (3.7; 50.6), 0.025

* Mann–Whitney U-test, ** linear regression model: DID (difference in difference), adjusted for day shift. ^1^ Significantly different from baseline, *p* = 0.001 (Wilcoxon test). ^2^ Significantly different from baseline, *p* = 0.001 (Wilcoxon test). ^3^ Significantly different from baseline, *p* < 0.001 (Wilcoxon test). ^4^ Significantly different from baseline, *p* = 0.008 (Wilcoxon test). ^†^ The toilet cubicle has a door; the toilet door can be opened (is unlocked); the toilet cubicle protects the user from outside views; the toilet door can be locked from the inside; the toilet bowl is intact. ^‡^ The toilet cubicle is free of litter; the toilet hole is unblocked; water for flushing is available inside the cubicle; the toilet bowl/pan/slab is free of visible traces of feces and urine; the cubicle walls and floor are free of visible traces of feces or urine.

## Data Availability

The dataset is available from the corresponding author on reasonable request.

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
