# Peer review of "Improving Toilet Usability and Cleanliness in Public Schools in the Philippines Using a Packaged Operation and Maintenance Intervention"

_ijerph, 2022, doi:10.3390/ijerph191610059_

Round 1

Reviewer 1 Report

The authors present a cluster-randomized controlled trial study to evaluate the impact of packaged interventions for O&M on the usability and cleanliness of toilets in public schools in Roxas city and Passi city in the Philippines. The paper is well written and the data analysis is statistically sound. I found the paper interesting and support publication with minor revisions.

1)      Line 57: ‘targets to achieve’ instead of ‘target to achieve’

2)      Table 1: Why does the mean of ‘total number of toilets’ not equal the means of ‘number of boys toilet’ + ’number of girls toilets’ + ’shared toilets’ for each group? Were there any exclusions, adjustments or outliers? Please clarify.

3)      Line 276: According to table, p = 0.002. Please check.

4)      Line 298: 19% Not 16%. Please check.

5)      Line 336: ‘supports girls attending school during menstruation.’ Reference needed

6)      Line 362: Why was the ‘toilet usability index’ not chosen for this study? Any strengths or weakness of this approach compared to the ones chosen in this study.

7)      Line 367-369: ‘provision of handwashing facilities and consumables (soap and cleaning supplies), budget allocation exercises, school-based management training, presence of a local champion, student and parental monitoring and involvement and support of the community are all effective components to improve sanitation O&M.‘ Did any of the schools in this study get additional support (or any interventions listed above) that may be responsible for increased usability or cleanliness?

8)      Lines 376-400: Honestly, without an O&M budget in the intervention package, a 4-month study period is very short for a conclusion on toilet usability and toilet cleanliness. I understand that COVID stopped the 8-month follow up for this study. However, some justification is required to support the argument that these interventions had/will have prolonged impact on WinS in Philippines.

Is there data from other interventions? If not, a qualitative check (possibly questionnaires or FGD or even personal communication) is needed from the intervention and control schools for usability or cleanliness?  

Author Response

The authors present a cluster-randomized controlled trial study to evaluate the impact of packaged interventions for O&M on the usability and cleanliness of toilets in public schools in Roxas city and Passi city in the Philippines. The paper is well written and the data analysis is statistically sound. I found the paper interesting and support publication with minor revisions.

We thank the reviewer for taking time to review our manuscript and for the valuable comments and suggestions.

  1. Line 57: ‘targets to achieve’ instead of ‘target to achieve’ 

We thank the reviewer for noting the spelling error. This has now been corrected.

  1. Table 1: Why does the mean of ‘total number of toilets’ not equal the means of ‘number of boys toilet’ + ’number of girls toilets’ + ’shared toilets’ for each group? Were there any exclusions, adjustments or outliers? Please clarify.

We checked the raw and cleaned dataset and noticed that for a few schools, the variable ‘shared toilet’ included values of ‘.’ (missing), which meant that there were zero shared toilets. These missing values were not included in the calculations of the mean number of shared toilets. This has now been corrected (see Table 1). This had no impact on the other analyses.

  1. Line 276: According to table, p = 0.002. Please check. 

We thank the reviewer for noting the error. This has now been corrected.

  1. Line 298: 19% Not 16%. Please check. 

We thank the reviewer for noting the error. This has now been corrected.

  1. Line 336: ‘supports girls attending school during menstruation.’ Reference needed

We have now added a reference for this statement (see reference 25).

  1. Line 362: Why was the ‘toilet usability index’ not chosen for this study? Any strengths or weakness of this approach compared to the ones chosen in this study. 

The reason we developed and used the Sanitation Assessment Tool (SAT) and the Cleaner Toilets, Brighter Future (CTBF) instrument, instead of the Toilet Usability Index (TUX) is that the TUX was a more complicated measure to use in the field (including many questions) and it demonstrated/measured little variation in toilet conditions in the Batangas Study. Also, the SAT was very much aligned with the JMP recommended core questions, making it a more valuable instrument for international use and comparison.

  1. Line 367-369: ‘provision of handwashing facilities and consumables (soap and cleaning supplies), budget allocation exercises, school-based management training, presence of a local champion, student and parental monitoring and involvement and support of the community are all effective components to improve sanitation O&M.‘ Did any of the schools in this study get additional support (or any interventions listed above) that may be responsible for increased usability or cleanliness?

The packaged O&M interventions this study also provided consumables, supplies and guidance for toilet maintenance, as well as tools to identify needed budget for repairs – in line with the previously listed interventions. Furthermore, the packaged O&M interventions were implemented in the already existing national WinS programme in the Philippines. In this programme, an infrastructure for capacity building, regular monitoring, provision of technical assistance by WinS coordinators at division level and community support is already in place. These aspects together likely contributed to the observed improvements in toilet conditions in this study. This information has now been added to the discussion section (lines 403-410).

  1. Lines 376-400: Honestly, without an O&M budget in the intervention package, a 4-month study period is very short for a conclusion on toilet usability and toilet cleanliness. I understand that COVID stopped the 8-month follow up for this study. However, some justification is required to support the argument that these interventions had/will have prolonged impact on WinS in Philippines.

We agree with the reviewer that four months is a short period and that we can therefore only draw conclusions on the impact of the interventions on the short term. We have now emphasized in our conclusions that positive results were observed over a period of four months, and that further research is needed to draw conclusions on the long-term impact of the packaged O&M interventions (see Discussion, line 344 and conclusion, lines 438-467).  

  1. Is there data from other interventions? If not, a qualitative check (possibly questionnaires or FGD or even personal communication) is needed from the intervention and control schools for usability or cleanliness?  

Unfortunately, we did not collect additional qualitative data during the study period, but we agree that this would be very important to collect in follow-up research to better understand the impact (or implementation aspects) of the packaged O&M interventions.

Reviewer 2 Report

Thank you for providing a well written paper on the evaluation of the intervention package to improve sanitation conditions in schools. 

I have minor questions that can be addressed to make the manuscript stronger.

a. How was the intervention package implemented, i.e. how often between baseline and follow up?

b. Was there any monitoring that was done between baseline and follow up? 

c. Please provide information on when the observations were done-i.e what time of the day, and how the time of observation influenced the results (if any)

d. Was there any influence of the number of students on the outcomes observed?

e. I notice that the observations were done by three evaluators. How was the evaluation of the three harmonised to one score? 

Author Response

Thank you for providing a well written paper on the evaluation of the intervention package to improve sanitation conditions in schools. 

I have minor questions that can be addressed to make the manuscript stronger.

We thank the reviewer for taking time to review our manuscript and for the valuable comments.

  1. How was the intervention package implemented, i.e. how often between baseline and follow up?

At the start of the study, school staff in intervention schools received an orientation training by the research team, during which they received the manual and materials with instructions on how to use them. They also received the MOOC video-based orientation training, produced by DepEd and GIZ, at baseline. After every two to three weeks, a member of the research team visited the schools to provide new consumables and to ask school staff if they needed any technical assistance. This information has now been added to the manuscript (see lines 151-159). 

  1. Was there any monitoring that was done between baseline and follow up? 

During the school visits by one of the research team members every two to three weeks, the research team member informally monitored the schools’ progress with implementing the packaged interventions and asked if any further assistance was needed. No formal monitoring assessments (e.g. interim data from questionnaires, observations or interviews) were conducted between baseline and follow-up. This has now been clarified in the manuscript (see lines 151-159).

  1. Please provide information on when the observations were done-i.e what time of the day, and how the time of observation influenced the results (if any)

We have now specified that data were collected during one day shift (either early morning, late morning, early afternoon or late afternoon) (see line 217). Since school toilets are usually cleaned in the mornings, we took the day shift into account in the analysis. Toilets were not significantly more clean in schools where toilets were assessed during the morning shift, but we decided to still adjust for the potential influence of the day shift in the regression analysis (see line 241-244).

  1. Was there any influence of the number of students on the outcomes observed?

Data on the exact number of enrolled students per school was not collected as part of this study. We only used the inclusion criterium that schools needed a minimum of 300 enrolled students to ensure enough toilets for assessment. Therefore, we are unfortunately unable to assess whether the number of students influenced the outcomes.

  1. I notice that the observations were done by three evaluators. How was the evaluation of the three harmonised to one score? 

During data collection, one examiner labelled the toilet, one examiner assessed the toilet using the SAT and one examiner using the CTBF instrument. These tasks were randomly rotated among the three examiners per school. Hence, no double assessments were performed, but examiners were trained and calibrated prior to data collection. Checks were performed at the end of each data collection round for consistencies between the number of toilets assessed using the SAT and CTBF tool. This information has now been further clarified in the manuscript (see lines 219-222).

Reviewer 3 Report

The manuscript “Improving toilet usability and cleanliness in public schools in the Philippines using a packaged operation and maintenance intervention”, by Denise Duijstera, Bella Monseb, Marvin Marquezc, Ubo Pakesd, Nicole Staufe and Habib Benzianf, presents results of the application of an operational package for the improvement of sanitary facilities (toilets) in schools. The package was tested in schools and positive results were observed.

In general, the manuscript is easy to understand and follows a logical sequence. The authors presented the relevant background for their work in the introduction. Information on the context of this study is also presented. The methodology (materials and methods) provides enough information on the experimental work. There is a good level of detail in the description of the methods as well as the experimental design and the statistical analysis that was applied.

I have two concrete comments/observations:

1) The water availability may not be the same for all schools. It is a well-known fact that access to water (quantity and quality) is the key factor for the success of many interventions intended to improve the sanitary condition. Did the author assess this crucial factor? The authors should include statements related to this aspect.

2) To assess the sanitation conditions, researchers prepared questions (to be answered) in concordance with the recommendations of international institutions. The authors could prepare an additional document (Supplementary Materials) in which questions and all material used for the assessment are presented. Many researchers reading this paper could be interested in these details. I also suggest the inclusion of data referred to the MOOC the authors claimed was used as part of the package. The data could be included within the paper or the Supplementary Materials. I guess the MOOC is still available and can be accessed by the readers. This is the reason for my suggestion.

According to the instructions for authors, Supplementary Materials can be “published online alongside the manuscript“. I think the authors should use this possibility since they have a lot of data of interest that can be published to support their paper.

In summary, this is a paper that may require the addition of Supplementary Materials. I am confident that the authors will complete the corrections in time considering the strict schedule of the journal. I believe that this paper can be considered for publication in the International Journal of Environmental Research and Public Health after minor revisions.

Author Response

We thank the reviewer for taking time to review our manuscript and for the valuable comments.

  1. The water availability may not be the same for all schools. It is a well-known fact that access to water (quantity and quality) is the key factor for the success of many interventions intended to improve the sanitary condition. Did the author assess this crucial factor? The authors should include statements related to this aspect.

We have now added information on the availability (and type) of water in schools. Availability of water was one of the inclusion criteria for the selection of schools (see Methods, line 133), and all schools met this inclusion criterium (line 248). Available water was often piped water, or a combination of piped water, rainwater or water from a deep-well. The water could not be used for drinking. This information has now been added to the Methods section, context (lines 109-110).

  1. To assess the sanitation conditions, researchers prepared questions (to be answered) in concordance with the recommendations of international institutions. The authors could prepare an additional document (Supplementary Materials) in which questions and all material used for the assessment are presented. Many researchers reading this paper could be interested in these details. I also suggest the inclusion of data referred to the MOOC the authors claimed was used as part of the package. The data could be included within the paper or the Supplementary Materials. I guess the MOOC is still available and can be accessed by the readers. This is the reason for my suggestion. According to the instructions for authors, Supplementary Materials can be “published online alongside the manuscript“. I think the authors should use this possibility since they have a lot of data of interest that can be published to support their paper.

We thank the editor for the useful suggestion. We have made the SAT and CTBF instruments available as supplementary material, as well as the manual and the video as part of the MOOC (see lines 160, 165-166).